# How Transformation Catalysts Take Catalytic Action

**Ju Young Lee** [1] and **Sandra Waddock** [2,*]

1  Ivey Business School, Western University, London, ON N6G 0N1, Canada; julee@ivey.ca
2  Carroll School of Management, Boston College, Chestnut Hill, MA 02467, USA
*  Correspondence: waddock@bc.edu; Tel.: +1-617-552-0477

**Abstract:** The challenges that are associated with the 17 United Nations sustainable development goals are wickedly complex and interconnected in nature. Because they require transformational changes at the systems level, the pace of change has, so far, been nowhere near fast enough to meet the goals by 2030. In this paper, we analyze the catalytic actions of a novel form of organizing that could potentially facilitate the timely achievement of transformational aspirations such as the SDGs: the transformation catalyst (TC). By identifying 27 TCs and analyzing their vision, mission, values, and their practices represented on their websites, we elaborate the following four key ways that TCs are distinctive from other entities, and therefore potentially more capable of facilitating transformational changes at the systems level: (1) TCs have transformation agendas that target systems-level solutions to bring about large-scale and fundamental changes in the relevant system(s), as opposed to more incremental or fragmented approaches; (2) TCs engage in catalytic actions, such as connecting, cohering, and amplifying the work of partners and collaborators; (3) TCs clearly acknowledge the current status quo, attributions, and urgency (i.e., sensemaking) of the issues on which they focus; and finally, (4) TCs embody systems orientation. In exploring how TCs work, we hope to build a solid conceptual framework for understanding the nature of transformative catalytic action on societal issues, and consolidate our understanding of what elements are needed if TCs are to work, providing a starting point for future research.

**Keywords:** transformation catalyst; catalytic action; system transformation; SDGs; connecting; cohering; amplifying; sensemaking

## 1. Introduction

In 2015, the United Nations set 17 sustainable development goals (i.e., UN SDGs) to be achieved by 2030. The SDGs seek to address complex societal challenges that scholars have often described as wicked problems [1], messes [2], or, more recently, grand challenges [3–5]. Such challenges include poverty, hunger, education, climate crisis, gender equality, water sanitation, biodiversity loss, and many more, all of which make achieving a better and more sustainable future for all difficult at best.

Admittedly, the UN SDGs have made significant inroads into the global business community and successfully elicited commitments from companies, as can be evidenced in the 2018 Oxfam report, which showed that 62% of companies have made a public commitment to the SDGs [6]. However, despite such seemingly high commitment and uptake, the pace of change is nowhere near fast enough to reach these goals by 2030. Unfortunately, the latest UNESCAP SDG progress report suggests that the Asia-Pacific region will likely miss all the goals by 2030. Further, the Bill and Melinda Gates Foundation, tracking inequality and progress made against the SDGs, based on 18 social indicators, also found, in their 2020 goalkeepers report, that the world has actually regressed on the vast majority of the global goals.

In a sense, this regression is not surprising. The nature of the challenges associated with the UN SDGs is so wickedly complex and interconnected, with conflicting interests of multiple actors, it is easy to reject any quick and easy solutions. The issues thus remain

stubbornly persistent in the nature of wicked problems [7]. Moreover, such wicked problems are fundamentally distinct from traditional problems, where there were clear problem definitions and easily traceable cause and effect relationships [1]. While traditional, 'tame' problems [1] can be solved in a linear manner, often by identifying best practices, wicked problems that manifest in complex environments reject such linear approaches [1,8]. In complex environments, where many relationships among elements are invisible, it is hard to clearly define what the problem is, and no single silver bullet can solve the problem, because the outcomes are often unpredictable and bring about unintended consequences [1,9,10].

To address these wicked problems in the increasingly interconnected, turbulent, and disruptive environment, a number of scholars have suggested a completely different approach from traditional models of innovation and problem solving, taking a systems perspective [9,11,12]. A system is defined as "an interconnected set of elements that is coherently organized in a way that achieves something" [13]. In the systems perspective, it is important to look at the whole, not just the parts, and focus on the relationships among the system's elements, which in living systems, such as organisms and human socio-economic systems, are dynamic, interactive, and nonlinear. A systemic perspective thus emphasizes how different systems elements are "interconnected and subject to non-linear, difficult-to-model dynamics because of feedbacks and delays" [8,11]. From this standpoint, systemic problems are messy and difficult to define, and the outcomes of actions are often unpredictable. Therefore, traditional, top-down models of intervention do not necessarily work or they can create unintended consequences [1]. In order to address complex and wicked problems, the systems perspective suggests nudging key "leverage points" within the system—"places within a complex system (a corporation, an economy, a living body, a city, an ecosystem) where a small shift in one thing can produce big changes in everything" [14].

In this paper, we focus on and elaborate on the role of a new form of organizing that embodies the systems perspective and potentially facilitates the timely achievement of SDGs by bringing about transformational changes at the systems level: the transformation catalyst (TC) [15]. TCs build on a legacy of other types of catalytic entities, such as catalytic alliances (CAs), global action networks (GANs), and field catalysts (FCs). We argue that TCs are distinctive in their explicit focus on systems that make them more capable of tackling wicked problems, such as sustainability and climate change, inequality, and other systemic issues, which are complex and interconnected in nature. While the previous catalytic organizations share a lot of roles in common with TCs, such as connecting, cohering, and amplifying different actors' efforts to achieve change, they did so in a rather siloed manner, whereas TCs not only connect and cohere actors, but also issues. They aim to make transformational changes at the systems level. Therefore, this paper is motivated by the following research question: how do transformation catalysts describe the catalytic work that they do?

To the best of our knowledge, this paper is the first empirical study of TCs. By identifying 27 TCs and analyzing their websites, we document how TCs are distinct from other entities, in their transformational agenda, catalytic actions, sensemaking, and systems orientation, all of which make TCs better suited for tackling complex problems and facilitating transformational changes at the systems level. We believe and hope that by providing new insights into what these new entities do and how they differ from their antecedents, our paper will contribute to bringing about the timely achievement of transformational aspirations such as the SDGs (though we do note the problematic nature of SDG 8, with its emphasis on continual growth on a finite planet).

## 2. Antecedents of Transformation Catalysts

Transformation catalysts have been defined as follows:

"Transformations catalysts (TCs) are promising organizing innovations specifically designed to address complexly wicked societal problems and opportunities and bring about purposeful system transformation. . . . Specifically, they connect,

cohere, and amplify efforts of other initiatives in an attempt to overcome the fragmentation and lack of impact . . . . They help coalitions of actors emerge shared visions, goals, aspirations, or other narratives that enable them to align their efforts, even while they pursue their individual agendas". [15] (p. 168)

In chemical reactions, the catalyst is an agent that brings about rapid changes without itself necessarily changing. In social circumstances, the idea of the catalyst has taken on a broader meaning of precipitating events or changes. In the case of TCs, those changes are transformative in nature, attempting to change the fundamentals of a situation [15].

TCs represent a new and still emerging way of organizing that is uniquely oriented to fostering transformational system change at a large scale [15], and that may offer some hope for dealing with the complexities with the wicked or even super wicked [16] problems associated with the UN SDGs and similar issues. We briefly explore TCs' predecessors below, before moving on to analyze how TCs view their own efforts in transformative or catalytic action.

In the 1980s, the idea of catalytic alliances was introduced [17] as catalytic social action, as follows: . . . catalytic social action involves the temporary alliance of organizations and their members to deal with an important problem . . . to foster longer-term change, without necessarily changing the nature or structure of the organizations in alliance (p. 394).

The social entrepreneurs who founded CAs used the following three mechanisms to engage others: a clear social vision "that has the potential to reshape public attitudes" (p. 394), significant personal credibility, and resources including networks, relationships, and followers' commitment to the collective purpose of the project. In 1995, the idea of catalytic alliances was elaborated, specifically identifying the characteristics of the CAs of the era.

Catalytic alliance initiatives operate at the leading edge of social reform, using the media as a strategic resource to place an issue on the public agenda and change public attitudes. Designed to leverage limited resources into major social change, by stimulating others (people and organizations) to take action, rather than intervening directly to affect a social problem, catalytic alliances work through an extensive network of other organizations to achieve indirect shifts in public attitudes. The objective is to spark a commitment to social action in others, through a high-profile media process, to move an issue onto the public policy action agenda, so that more traditional organizations (public, private, nonprofit) can effectively work on the problem [18] (pp. 951–952).

More-complex forms of networks that also emphasize the (catalytic) transformation of a societal issue are referred to as issue networks [19], global action networks (GANs) [20], and global governance organizations [21]. GANs are multi-stakeholder global networks that tend to focus on particular issues. For example, the Global Reporting Initiative (GRI; environmental, social, and governance reporting for organizations), the Forest Stewardship Council (FSC; forest sustainability), and the Global Water Partnership (GWP; water security) are all GANs. GANs share some TC characteristics, but generally limit their actions to particular approaches to solving their focal issue (e.g., corporate reporting, forest certification, water resource management). They take direct action to change the issue through everything from local- or industry-based certification, through national–regional–global processes to create supportive policy and market dynamics.

In 2018, Hussein and colleagues introduced the idea of the field catalyst (FC), which, similarly to CAs and GANs, "sought to help multiple actors achieve a shared, sweeping goal" [22] (p. 50). The authors argued that such FCs shared the following four characteristics: emphasis on "achieving population-level change" by building or strengthening a field, "influenc[ing] the direct actions of others, rather than acting themselves", focusing on getting things done (not achieving consensus), and being "built to win, not to last" (p. 50). Hussein and co-authors, similarly to others before them, noted the importance of the vision/mission. They indicated that the vision needed to be "big enough for stakeholders to rally around, yet specific enough to make a measurable difference" through a "road map

for change" by, in a sense, connecting the dots among enterprises already engaged on the relevant issue or in the field (p. 51).

There are other antecedents to TCs. For example, innovation brokers or intermediaries work, typically in the context of community-based initiatives (CBIs), to develop and maintain networks of similarly minded organizations, by creating an interstitial (between organizations) infrastructure [23] (see [24] for types of such entities). This infrastructure enhances the capacity of CBIs to network, collaborate, and learn from each other, as well as to engage in policy activism [23]. CBIs perform a type of curatorial role that is necessary for navigating complexity that helps connect previously unlinked actors in a catalytic way and work to ensure that knowledge is shared by diverse actors across multiple scales [25].

Innovation brokers or intermediaries serve useful functions by creating networks of innovators, working collaboratively around similar visions, in ways that enhance support for innovations, and working across multiple boundaries [26,27]. Similarly to TCs, innovation brokers emphasize the importance of a good story or narrative that compels relevant actors to take action and taps particular windows of opportunity that exist across different types of innovators [27]. Unlike innovation brokers, however, TCs go beyond innovation systems and facilitate multi-stakeholder innovation, and magnify efforts [28].

Catalytic change also features prominently in Christensen's notion of catalytic innovation for social change [29]. In Christensen's framing, catalytic change is explicitly oriented to product/market innovations by companies, rather than the societally or systems-oriented actions of TCs. Though such innovations may serve social purposes, and bring about significant changes in different product/market arenas, catalytic innovations, in Christensen's framing, are firmly based in markets, rather than attempting the broader systemic transformation of catalysis at which TC's aim.

Looking at a related structure, Selsky and Parker [30] (p. 850) synthesized the literature and defined cross-sector social partnerships (CSSPs) as "cross-sector projects formed explicitly to address social issues and causes that actively engage the partners on an ongoing basis". Generally speaking, CSSPs directly deal with specific issues, rather than working catalytically to cohere or align other actors who are dealing with relevant issues, to amplify their impact. In other words, while they are cross-sector and relationally based, they directly work on whatever the relevant issues are, though none are necessarily oriented towards catalytic or transformative system change as TCs are.

The literature on CSSPs also highlights the role of partnership brokers [31–33]. Although partnership brokers, similarly to TCs, convene potential partners and consolidate robust working relationships [33], working "to steer and support the building of partnerships" [31] (p. 2), they also differ from TCs in a number of important ways. First, partnership brokers can be internal (i.e., from within one of the partner organizations) or external (i.e., independent of the partner organizations), whereas TCs are solely external, in that they do not take direct actions themselves, but instead focus solely on connecting, cohering, and amplifying works of other actors. Secondly, the changes that partnership brokers intermediate are often issue-specific and not necessarily transformational in nature.

## 3. Identifying What Transformation Catalysts Do

TCs have emerged in two interconnected contexts. First is a growing recognition of the need for systemic transformation change, identified by many observers, especially in the wake of the COVID-19 pandemic, and as articulated in the SDGs and the climate emergency. Understanding this context combines with the awareness of the complexly wicked nature of the systems that need to change, which means that no single actor will be able to bring about transformation [34]. The other context has a growing capacity to work catalytically with a variety of actors, often online, connecting efforts that are attempting such transformational change in new ways, to bring about more effective, situationally relevant, and impactful transformation of whole systems at different levels of analysis. The emergence of Zoom and similar platforms has enabled new types of connections and

alliances that might not have been feasible in a less digitally connected era, or where people were still able to get together easily in person.

The rest of this exploratory paper analyzes and synthesizes what the 27 entities that have been identified as TCs articulate on their websites about what their work is and how they approach it. In exploring these TCs' work, we hope to build a solid conceptual framework that helps change agents to better understand the nature of transformative catalytic action on societal issues.

*Research Orientation and Methods*

TCs are a relatively new organizational phenomenon, hence little is known about how they define and implement their catalytic work. Because this paper is exploratory in nature and descriptive in its results, both the initial identification of TCs and the findings must be considered to be tentative. We focused on the following research question: how do transformation catalysts describe the catalytic work that they do? Using their descriptions, we hoped to be able to determine what is meant by catalytic action in the context of the transformation catalyst. As described below, data about their activities and aspirations were collected from candidate TC websites, using the aspirational statements on their webpage, as well as statements associated with their vision, mission, values, "what we do", "about" pages, or similar, to provide as much insight as possible into their activities.

The data were gathered into individual Word documents for each entity and input into the NVivo program for coding. Initially, we drew from the Transformations Catalyst paper mentioned earlier, to develop tentative coding categories, then the two authors separately coded data from about a quarter of the websites, to see what additional categories emerged. After consultation with each other, we developed a common framework with four major categories (transformation agenda, catalytic action, sensemaking, and systems orientation), with relevant subcategories, as described in the next section (see coding in Figure 1).

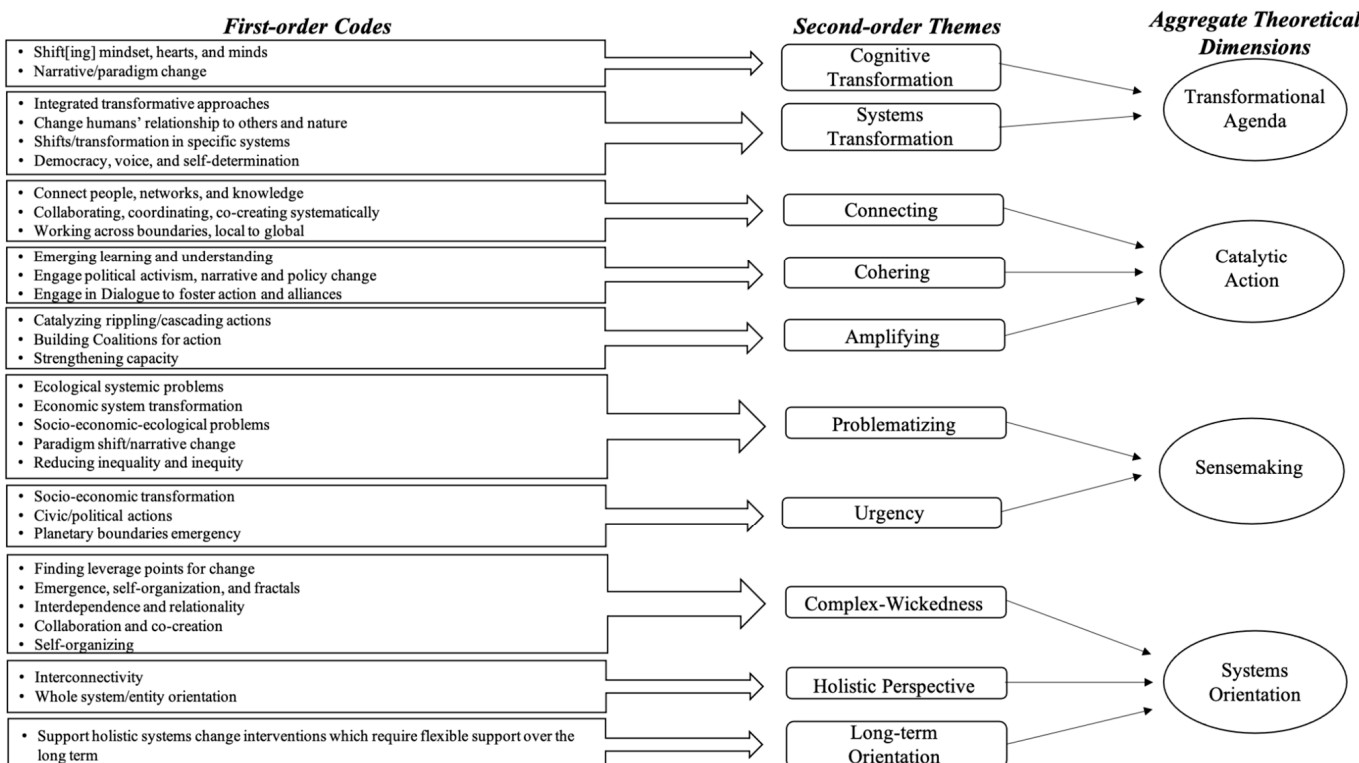

**Figure 1.** Coding scheme for transformation catalyst action strategy data.

More specifically, developing these criteria was actually part of an iterative, inductive (and admittedly somewhat circular) research process that enabled us to both define and

discover TCs, and also pointed us in the direction of appropriate codes later on. One of the authors had been working intensively with several organizations that were clearly identifiable as TCs and had some initial understanding of what we were looking for. Using a version of the general definition of TCs above and exploring a number of possible candidates, we went through an iterative process of exploring different entities' websites. In doing so, we developed the following four main qualifying criteria for potential TCs (see details in Supplementary Material S1): transformation agenda; catalytic action of some sort, sensemaking, and systems orientation. One of the authors coded about a quarter of the TCs candidates using the working list, adding new ideas and details as they emerged from the data. Each of these criteria became the main code category, with final subcodes emerging from the data itself in an iterative process. We solidified the criteria once saturation was reached and no new criteria were emerging (also eliminating any existing criteria that did not seem to be present in the data). Once the codes were established (see Figure 1), the other author similarly went through the same 25% of TC candidates as a validity and reliability check, then finished coding the rest of the data for all of the TCs (note: all Supplementary Materials are online).

This process enabled us to tentatively identify 27 entities as TCs. Of the initial 41 candidates identified, 14 were eliminated as they did not meet these criteria. We then gathered rich, qualitative data from the websites of all the possible TCs, i.e., information that described their vision, mission, "how we work", "about us", and related categories, during October 2020. The data revealed the following four main sets of activities, most of which had subcodes, which will be discussed in detail in the next section: transformation agenda (what and where), systems orientation (approach), catalytic action (who and how), and sensemaking (why and when). Once all the data were coded, we began to analyze the data, starting by developing both word clouds and Excel spreadsheets that indicate the top words used in each code, to get a sense of how each category was framed and what types of activities were involved. Then, we created documents of key phrases from each code category that provided more in-depth, qualitative rounding to each of the codes. This latter step enabled us to synthesize each of the codes and subcodes, providing the needed insight into how these TCs take action and what they mean by catalytic actions, as observed in Table 1 and discussed below.

**Table 1.** Transformation catalysts' views of catalytic action.

| Overarching Action | Defined as | Evidenced by |
|---|---|---|
| The 'What' and 'Where': systems transformation agenda: | Systems transformation involves targeting systems-level (or whole-system) solutions to bring about large-scale and fundamental changes in the relevant system(s) versus more incremental or fragmented approaches | ***Cognitive transformation***: bringing about shifts in peoples' mindsets, mental models, and paradigms by reconfiguring and transforming what are known as cultural narratives and telling inspiring new stories<br>• Mindset change (mental models, paradigms)<br>• Narrative/paradigm/cultural mythology change (and underlying memes)<br><br>***Systems transformation***: targeting systems-level (or whole-system) solutions to bring about large-scale and fundamental changes in the relevant system(s) versus more incremental or fragmented approaches<br>• Integrated transformative approaches<br>• Changing humans' relationship to others and nature<br>• Shifting/transforming specific systems<br>• Enhancing democracy, voice, and self-determination |

| Overarching Action | Defined as | Evidenced by |
|---|---|---|
| The 'Who' and 'How': catalytic actions | Catalytic actions involve connecting, cohering, and amplifying the work of partners and collaborators, defined as bringing together a network of change-makers and supporting collaboration across, disciplines, sectors, nations, and other boundaries to co-create and emerge transformative change and build sustainable futures for all | *Connecting*: connecting initiatives/people together to inspire them to collaborate, coordinate, and co-create systemic action in the desired direction.<br>• Connecting people, networks, and knowledge<br>• Collaborating, coordinating, co-creating systemically<br>• Working across boundaries<br>*Cohering*: building strong alliances and collaborative relationships across silos by combining, unifying, and synthesizing knowledge and strategies that build capacity to act and finance transformative change<br>• Emerging learning and understanding<br>• Engaging political activism, narrative, and policy change<br>• Engaging in dialogue to foster action and alliances<br>*Amplifying*: strengthening and empowering diverse groups of actors to organize, mobilize, and take action to create transformative change that impacts at different levels (community to regional to national and global)<br>• Catalyzing rippling/cascading action<br>• Building coalitions for action<br>• Strengthening capacity |
| The 'Why' and 'When': sensemaking | Sensemaking involves TCs clearly acknowledging why and when transformative change is needed in a broad variety of contexts, recognizing issues and their impacts, and articulating/disseminating the urgent need for transformative change and how it will be done, including shifting narratives. | *Problematizing specific topics*: articulating the problems in today's systems and the sometimes existential challenges to humanity that they represent, and arguing for a paradigm or systemic transformation towards flourishing futures<br>• Ecological system problems<br>• Economic system issues<br>• Socio-economic-ecological problems (integrated)<br>• Paradigm shift/narrative change<br>• Reducing inequality and inequity<br>*Urgency for transformation*: acknowledge that addressing the problems raised (in problematizing) requires urgent systems-level transformation at speed and scale that can only be achieved through targeted actions and mobilizations<br>• Socio-economic transformation<br>• Civic/political actions<br>• Planetary boundaries emergency |
| The 'approach': systems orientation | Adopting a systems orientation (systems thinking), which means thinking in terms of complex adaptive systems and wicked problems (with or without that specific language) and taking a holistic perspective on systemic change | *Systems orientation*: adopting a systems understanding (systems thinking), which means thinking in terms of complex adaptive systems and wicked problems (with or without that specific language) and taking a holistic perspective on systemic change<br>*Complex wickedness (wicked complexity)*<br>• Finding leverage points for change<br>• Emergence, self-organization and fractals<br>• Interdependence and relationality<br>• Collaboration and co-creation<br>• Self-organizing<br>*Holistic perspective*: recognizing that everything in complex wickedness is interconnected, spans multiple levels and sectors, and therefore need to be tackled holistically rather than in silos because the systems of interest can be considered living systems<br>• Interconnectivity<br>• Whole-system/entity orientation<br>*Long-term orientation*: seeing things and the prospects of systems over the long term, seeking long-lasting solutions and changes in systems to achieve long-term environmental and social sustainability for the future |

## 4. Results

As discussed above, there are four main categories of coding, all of which have subcodes that help distinguish specific activities within that category. Combining the top words with the qualitative data listed in NVivo references (compilations of data coded in different categories) enabled us to define each of the codes and provide insight into the specifics of how the TCs themselves view each of the following activities associated with catalyzing transformation: developing a transformation agenda, taking a systems orientation, taking catalytic action, and sensemaking.

Below, we discuss each of these categories of catalytic action/orientation. Here, we are interested in how entities identified as transformation catalysts actually undertake their work—in their own words—and whether we can derive commonalities across such entities that help to understand them better. With that understanding, it becomes possible for actors who are interested in system transformations towards flourishing to emerge new TCs as needed, consolidating the understanding of what elements are needed if they are to work. Table 1 synthesizes the overall findings.

### 4.1. Transformation Agenda: The "What" and "Where"

The transformation agenda of TCs is generally defined as *what* TCs aim to transform and *where* they attempt to do it. TCs transformation agenda involves the adoption of a systems perspective on where transformation is needed, acknowledging places or systems and subsystems that need to shift and cognitive (mindset and narrative) transformations that are needed. Two primary aspects of *what* and *where* emerged from the data of cognitive transformation and systems transformation. In other words, change needs to happen with respect to mindsets or paradigms that either hinder or support transformation, and change needs to happen at the systems level, not in a piecemeal or atomized fashion (see Supplementary Material S2 for details and Table 1 for a synthesis).

Picking up on Meadows' [14] insight that mindset change is the most important change lever either implicitly or explicitly, many TCs emphasize shifting narratives as an underpinning of mindset change. They also, importantly and not surprisingly, focus on transformative change achieved through systems transformation. Looking at the listing of the top words for all transformation agenda codes (see Supplementary Material S2) highlights the emphasis on transforming, developing, building, and particularly changing systems, particularly economic systems, towards sustainability, and a world that works for humans and communities. Other words that appear most often in this overall category focus on the narrative and actions, such as building, creating, as well as the idea of a living planet. As noted, however, the transformation agenda has two main aspects, which we labeled cognitive transformation (mindset change, as Meadows would call it) and systemic transformation or change, both of which we explore in some depth below.

### 4.1.1. Cognitive Transformation

One aspect of *what* and *where* of the transformation agenda is oriented towards is cognitive transformation. Cognitive transformation aims to bring about shifts in peoples' mindsets, mental models, and paradigms, by reconfiguring and transforming cultural narratives and telling inspiring new stories. Cultural narratives are important because they serve as core mythologies [35] that tell people how the world works and what their place in it is. That is why Meadows places so much importance on mindsets and the need to shift paradigms as leverage points for systemic change [14]. Mindsets and paradigms, comprised of core ideas or cultural units that are sometimes called memes (ideas, symbols, images, and words), influence attitudes, behaviors, and practices [36,37]. Supplementary Material S2 top word listing for cognitive transformation highlights the importance of narrative change for transformative impact, with an emphasis on emerging or developing new human and economic systems, knowledge, and impacts.

Insight about how TCs frame this mindset or cognitive transformation can be gained by looking more in-depth at how they describe it. There are two aspects of what we have

labeled cognitive transformation reflected in the data, as follows: mindset change, and narrative or paradigm change (Supplementary Material S2 provides sample quotes for both aspects, which are synthesized briefly below).

*Mindset Change*. A core aspect of the transformation agenda is shifting mindsets, mental models, or paradigms (belief systems). Mindset change, as Meadows argues, has to do with how people perceive and relate to the world around them, and what their mental model or paradigm is of that world, other people, and, for example, human relationships with nature [14]. This fundamental aspect of the transformation agenda is encapsulated by one TC as a need for "shifting, reconfiguring, and transformation; mindsets, mental models, and paradigms, through intentional processes and design, purposeful interventions, and conscious, deliberate approaches". Another points to the underlying rationale for such change as "shift[ing] hearts and minds" and the recognition of the "potential of narrative as a catalyst for transformation.

*Narrative/Paradigm Change*. The other core aspect of the transformation agenda is narrative or paradigm change, which obviously is associated with a mindset change, and represents the actual shaping of a new paradigm. As one TC puts it in discussing its agenda, it wants to "help the global community develop inspirational visions and stories . . . to shape the very reality that they forecast". Another provides the core rationale for narrative/paradigm change, as follows: "we need stories about the future that paint a realistic and optimistic vision of what the world can be". The idea of paradigm change reflects Buckminster Fuller's keen and well-known insight that "you never change things by fighting against the existing reality. To change something, build a new model that makes the old model obsolete".

### 4.1.2. Systems Transformation

The second main aspect of the what and where of the transformation agenda is targeting systems-level solutions in bringing about large-scale changes and transformations, instead of trying to solve individual challenges in silos in an incremental manner. Systems transformation, as evidenced in this code, involves targeting systems-level (or whole-system) solutions to bring about large-scale and fundamental changes in the relevant system(s) versus more incremental or fragmented approaches. The most used words in the system transformation code (see Supplementary Material S2) indicate that the what and where of systems transformation is transforming and changing (particularly) economic systems globally towards sustainability and a world that works for all people/humans, where climate change is controlled. There are the following four main ways in which these what and where aspects of systems transformation are expressed: an integrative set of approaches encompassing multiple systems (integrated transformative approaches), changing humans' relationships to others and nature, shifts/transformation in specific systems, and enhancing democracy, voice, and self-determination. Supplementary Material S2 provides sample quotes for each, which are briefly summarized below.

*Integrated Transformative Approaches*. Taking integrated or comprehensive approaches to the need for system transformation means, as one TC puts it, means "advocat[ing] for paradigm and systems shifts which will enable society to emerge from our current crises by promoting a new way of being human within a more resilient biosphere". Such approaches will be multi-pronged and encompass numerous aspects of system change, including narrative change, and recognition of the complexity, interdependence, and interconnectedness of systemic challenges, as will also be discussed below, in looking at the transformations agenda.

One TC's idea encapsulates this complexity and need for holistic approaches quite well, with implicit recognition of the massive scope and many different moving parts of system change. This TC says it will "facilitate systemic change interventions at local-to-global levels, working partnership with various stakeholders . . . : patterns, address underlying root causes, engage in the potential of living systems, solve big social issues, with the outcome of creating, enduring and positively affecting different behaviors and

outcomes, resilient, lasting and better results, building a bridge to a better tomorrow, increased systems health, positive social change, just sustainable, and compassionate societies, a new normal, the emergence of a new system and a new reality".

*Changing Humans' Relationships to Others and Nature.* Other TCs explicitly recognize that humans need to relearn that we are intimately connected with, and dependent upon, nature. This recognition, which draws on ancient Indigenous wisdom, argues that a flourishing future for all means building on both that recognition and new behaviors and practices that acknowledge that interdependence. For example, one TC summarizes this perspective as "true wealth derives from humans in harmony with a healthy biosphere". Another TC expresses this aspect of transformation as regenerative, while another expresses it as "transformation of humanity's activities so they restore nature, rather than continue to destroy it".

*Shifts/Transformation in Specific Systems.* Some TCs focus on specific systemic issues or subsystems as a way of bringing about the change that is needed, by finding what Meadows called key leverage points [14]. One such leverage point is the financial system, which currently drives the way economics and business are conducted in the world. One TC calls for "a paradigm shift across mainstream finance so that it acts to restore rather than destroy nature". In a related theme, another TC "reimagines our economic and financial system to promote a transformation to a more just and regenerative world". Others focus on the policy environment, arguing "improved legal and regulatory environments", or a "vision of sustainable, equitable economies that work for all". The economic (and finance) systems are at the core of the envisioned changes in this thread, in what one TC describes as seeking to "address political-economic rules change head-on" to achieve those economies.

*Democracy, Voice, and Self-Determination.* While not as often expressed as the above themes, the core idea of self-determination and voice does appear in the efforts of many TCs. They, for example, argue that communities should be "self-determined: ... have the power to own, transform and create wealth in local economies that can best serve their interests". Another TC seeks "being true to fundamental values regarding human freedom, dignity, respect, equal opportunities, and human rights".

*4.2. Catalytic Actions: The "Who" and "How"*

Catalytic action is at the heart of the transformation catalyst's work, as it details the who and the how of transformational work for these entities. In this section, we explore the following three main themes that emerged from the data related to catalytic action: connecting, cohering, and amplifying the work of their partners and collaborators, for more effective action and a greater impact, each of which has multiple subthemes, as discussed below. Based on the data, catalytic action can be defined as bringing together a network of change-makers, and supporting collaboration across, disciplines, sectors, nations, and other boundaries to co-create and emerge transformative change and build sustainable futures for all. That is the who of transformation catalysis. Table 1 shows this definition, most used words/memes in this category (see Supplementary Material S3 for details), and sample quotes for each of the subthemes.

4.2.1. Connecting

Connecting means bringing together a group or network of change-makers and supporting collaboration across that network, including across disciplines, sectors, nations, and other boundaries, to co-create and emerge transformative change and build sustainable futures. The data reveal the following three themes associated with connecting: connect people, networks and knowledge; collaborating, coordinating, co-creating systemically; working across boundaries—local to global. We will discuss these themes briefly below, and further examples from the coded data are included in Supplementary Material S3.

*Connecting People, Networks, and Knowledge.* The idea of connecting people, networks, and knowledge is core to the connecting function and an important aspect of the "how" of the catalyzing action. Building on the fragmentation that is evident in what Hawken [38]

called "blessed unrest", or the existence of numerous small and unconnected entities generally working towards an equitable and ecologically flourishing world, the fundamental idea behind connecting is to diminish the blessed unrest and build connections among these initiatives. The hoped-for result of bringing these entities together—connecting them—is to build strength in numbers and, as expressed in the theme of amplifying discussed below, to enhance the impact. This idea is expressed succinctly by one of the TCs as "brings together people and organizations, ready to seize the opportunity for creating a sustainable transition towards a more human, equitable, and flourishing world". The knowledge generation aspect of that is expressed by one TC as "make global knowledge and democracy more available by supporting inter/transdisciplinary dialogue and connectivity", while another wants to develop "an ecosystem of stories that . . . will inspire others to join". The core element of this theme is to "(re) build . . . an economy that includes everyone".

*Collaborating, Coordinating, Co-Creating Systemically.* Simply connecting initiatives or people together, however, is insufficient to create the impact that is needed. Another important aspect of the connecting activity of TCs is to inspire connected entities to collaborate, coordinate, and co-create actions that impact the systems of interest. This need and desire is expressed in a number of ways. One TC says that it hopes to "build and mobilize networks", while another attempts to "coordinate action: . . . create the strategies, processes and platforms needed to build strong and genuine alliances, across silos". The notion of reducing fragmentation and silos is important here and impacts both the connected initiatives themselves and the world that they aim to build. One TC neatly summarizes this theme in the following way: 'We are bridge builders and connectors: people committed to moving beyond the confines of 'us vs. them' to build a world where everyone belongs".

*Working Across Boundaries—Local to Global.* The key to connecting as catalysis is that actors are connected across a variety of boundaries, which might be disciplinary, sector, issues, bioregional, or bounded by some sort of government, i.e., city or town, province or state, national, or even global. The need for the reduction in silos and associated fragmentation of work and, therefore, impact is expressed by the TCs as, in one case, the need to "catalyze connections between governments, research institutions, businesses, civil society, and leaders, to maximize the impact of everyone's efforts". A similar thought is expressed by one TC to "connect the small groups to each other and the larger whole". This theme is expressed fully by one TC as "building a global, cross-sector network via partners . . . to lead efforts in regions across the global consistent with the priorities appropriate to their context". Note here that the idea of voice at the appropriate level is implicit in this way of operating, as is the recognition that systems are complex and that boundaries must be crossed if effective transformative action is to be taken.

### 4.2.2. Cohering

The second form of catalytic action is labeled as cohering. In the TC context, cohering means building strong alliances and collaborative relationships across silos, by combining, unifying, and synthesizing knowledge and strategies that build the capacity to act and finance transformative change (see Supplementary Material S3 for details, and the summary in Table 1). As a catalytic activity, cohering the work of numerous initiatives has the following three dimensions (or themes): emerge learning and understanding; engage political activism, narrative, and policy change; and engage in dialogue to foster action and alliances.

*Emerge Learning and Understanding.* Coherence has to do with the ability to create and tell a new story—or narrative—and in order to do that, TCs need to bring together many different aspects of that knowledge and shape it in ways that others can understand. Part of that task appears in the related catalytic activity of sensemaking, as discussed below, but the cohering activity is one of synthesis—either of knowledge and understanding or of actors themselves. The knowledge synthesis aspect of emerging learning and understanding is expressed clearly by one TC, as follows: "synthesize and disseminate knowledge and evidence about what . . . looks like and how we get there facilitating the

difficult conversations and the bold actions required to confront the planetary emergency facing humanity and our common home". Creating coherence among actors or partners is expressed by another TC as follows: "convene, facilitate, and foster collaboration amongst actors and actions that hold a shared interest in promoting and developing responsible leadership". Other TCs bring both of these synthesizing activities together, as in the following statement: "unifying: science, insight, and implementation *with* stakeholders" and, another "convenings for knowledge, interaction, action", which is the central point of cohering knowledge and understanding.

*Engage Political Activism, Narrative, and Policy Change.* Many TCs recognize the need to impact policy, as well as high-level narrative change, in order to bring about the transformations that they believe are needed. This theme is most evident in the following types of statements: "to activate political-economic rule change, civic mobilization, and narrative change . . . the activation of these to catalyze change". Policy change is, for many TCs, a vital element of the change that is needed, even when that change is emerging from the ground up, because policy and narratives create the ecosystem or context in which change needs to occur.

*Engage in Dialogue to Foster Action and Alliances.* Another theme that emerges in the cohering category of catalytic action is engaging partners, collaborators, and other actors in dialogic processes that have the outcome of fostering greater action. One TC summarizes this aspect as acting to "bring our community together to engage in meaningful dialogue to advance best practice". Another emphasizes coordinating actions that "build strong and genuine alliances across silos", echoing the earlier themes of coherence. This notion can be synthesized by one TC's statement that it is "exploring how to work together to better intervene in systems".

### 4.2.3. Amplifying

The amplifying aspect of catalytic action involves strengthening and empowering diverse groups of actors to organize, mobilize, and take action, to create transformative change that impacts at different levels (community to regional to national and global). In a sense, it is the amplifying activity where the catalysis that begins with connecting and cohering, discussed above, really gains its effectiveness. Amplifying, as evidenced in the data, has the following three aspects or themes that are explored below: catalyzing rippling/cascading actions; building coalitions for action; and strengthening capacity (see Supplementary Material S3 and summarized in Table 1).

*Catalyzing Rippling/Cascading Actions.* The idea of creating ripple or cascade effects is central to the notion of catalysis. Thus, TCs variously express the need to "create big ripples" or "unlock chains of cascading ecological, social, and economic benefits". One TC uses the word amplifying to express its intent, as follows: "amplifying voices—celebrate and lift up ideas, solutions, and diverse voices that aren't heard". As noted earlier, a chemical catalyst creates big changes without necessarily changing itself, but, in social contexts, catalysts are part of the system that they are attempting to transform. Getting to real transformative change, however, means that incremental actions, or actions without significant leverage (as Meadows would say), will likely not achieve desired results, hence the need—and recognized capacity by TCs—to create ripples from their work.

*Building Coalitions for Action.* Another important aspect of amplifying TCs impacts comes from building coalitions that are organized for action and impact. While this theme is related to that of connecting, the emphasis here is on action and impacts, stated by one TC quite explicitly as "build coalitions and catalyze change". Another TC elaborates that it is building "coalitions of remarkable leaders who transform business sectors, societies and economies, nationally and globally". The overall intent of such coalitions is synthesized by the following TC: "brings together real economy actors, investors, thought leaders to address core systemic alterations needed to support a well-being economy in balance with nature and responding to core global tipping points".

*Strengthening Capacity*. Clearly, amplifying means strengthening capacity in a variety of ways, though, as one TC calls it, "network weaving" or, another, "work[ing] in partnership, and working against power dynamics". Some TCs explicitly mention the idea of strengthening capacity. For example, one TC expresses this theme as "strengthening capacity and processes to engage". Another synthesizes this theme as follows: "strengthening the power of people to organize, mobilize, take action".

*4.3. Sensemaking: The "Why" and "When"*

Sensemaking of TCs involves TCs clearly acknowledging why and when transformative change is needed, in a broad variety of contexts, especially the current socio-economic system, recognizing issues and their impacts, and articulating/disseminating the urgent need for transformative change and how it will be done, including shifting narratives. There are the following two core aspects of sensemaking, as framed by TCs in this study: problematizing and urgency (see Supplementary Material S4 and synthesis in Table 1). TCs need to provide a rationale for conducting the catalyzing work, and the sensemaking activity is how they do that. As we note, there are multiple themes associated with both problematizing and urgency, as discussed below.

4.3.1. Problematizing

Problematizing by TCs can be defined as articulating the problems in different domains or specific topical areas in today's systems, and the sometimes existential challenges to humanity that they represent, and arguing for a paradigm or systemic transformation towards flourishing futures (see Supplementary Material S4). Themes in problematizing argue mainly for why transformation is needed. Though they are clearly interrelated and somewhat overlapping, here we sort them into the following categories because they highlight slightly different aspects of the broad context of problems that seem to demand transformation: ecological systemic problems; economic system transformation; socio-ecological problems (combining the first two); paradigm shift/narrative change; and reducing inequalities and inequity.

*Ecological systemic problems*. The theme of ecological systemic problems focuses on issues associated with the natural environment, including the interrelated issues of climate change and biodiversity loss, as well as negative human impacts on nature. The three quotes in Supplementary Material S4 illustrate these emphases. One TC argues for the need to "avert climate breakdown and biodiversity collapse and begin to put in place the resilient and flourishing systems required for all life on earth to continue". Another articulates the issues as a planetary emergency, noting that "climate change and biodiversity loss are today's most pressing global challenges, constituting an existential threat to humanity". A third TC raises similar issues, pointing out that "continuing exploitive and extractive human activities on the planet combined with climate change pose a truly existentialist question to society". This form of problematizing raises the question of the very continuation of the human species or at least civilization as it is known today.

*Economic system transformation*. Perhaps the most common theme associated with problematizing is that of economic system transformation, which many of the TCs view as a core problem that has created the ecological issues that were just discussed, as well as the ones discussed below. One TC states this framing very baldly, as "our economic system is broken. It's time to reset". Another, largely agreeing, states, "capitalism is broken. Too many people have been left out". A more nuanced assessment of the same issue of economic system transformation is given by other TCs, with one highlighting that "the paradigms and values that underpin our current political-economic system, and the rules, laws, and conventions that maintain these, are the root causes of humanity's imbalance with nature", and another discussing the "fundamentally flawed economic theory and indicators". One TC emphasizes the "relentless and narrow pursuit of infinite growth . . . without any grounding in moral and ethical values" as being at the heart of the problem it

is trying to deal with. This theme consistently pushes towards a reframed and transformed set of economic principles and values as a key to transformation.

*Socio-Economic-Ecological problems.* An even more complex approach that draws from the first two themes within the category of problematizing identifies socio-economic-ecological problems, i.e., the combination of these issues, as what needs to transform—suggesting the scope of transformation that may be needed for bringing about a flourishing world for all. That notion can be simply stated, as one TC does, by "address[ing] the multiple crises facing humanity and the planet", while another synthesizes it by commenting that the "global socio-ecological challenges demand a new context". The variety of problems that need to be reckoned with, at least in one TC's view, is broad and complex, including inequality, a growing human ecological footprint that threatens natural systems, with the potential for instability and conflict.

*Paradigm Shift/Narrative Change.* Narrative change, as part of mindset change, has been discussed above, and it is also part of the problematizing category when the current dominant narrative is discussed. When economics and many ecological issues are discussed, the narrative tends to focus on the problems associated with neoliberalism with its known socio-ecological impacts [39,40].

*Reducing Inequality and Inequity.* When TCs focus on specific issues within the systems context of interest, they frequently land upon issues of inequality and inequity, which Hawken (2007) too found to be central (along with sustainability) to the work of the many initiatives operating in "blessed unrest". Inequity here also includes racial injustice, especially as the data were gathered after protests in the U.S. about racial profiling by police and systemic injustice. One TC paints that picture openly, "economic equity will be achieved when we talk explicitly about race and the wealth gap". Another argues that "the current moment of a global call to action around racial justice, set against the backdrop of a global pandemic ... raises the stakes in what was already an era of intersecting crises", which include economic and social inequality among others.

### 4.3.2. Urgency

Many of the TCs paint a picture of urgency around their catalytic missions. Here, urgency means acknowledging that addressing the problems raised (in problematizing) requires urgent systems-level transformation at speed and scale that can only be achieved through targeted actions and mobilizations (Supplementary Material S4). Three frames that forwarded this urgency emerged in the data, as follows: socio-economic transformation, civic/political actions; and planetary boundaries emergency, which, of course, reflect themes already discussed above.

*Socio-Economic Transformation.* The socio-economic transformation agenda is a clear point of urgency. One TC puts it boldly in the following: "an economic transformation that promotes human flourishing for all on a thriving planet is irrefutably and urgently needed". In a similar vein, another comments, "nothing less than a transformation in the political-economic rules that govern humanity's relationship with nature will turn around the present trends and begin rebuilding a robust and diverse nature".

*Civic/Political Actions.* Another way of expressing the urgency of the moment is to argue that "collective action and bold, new strategies are needed to address the crises of our time". One TC argues for similarly bold action via "strategic and targeted civic mobilization, enabled by narrative change". Moreover, one TC states "we are in dire need of a paradigm shift and an upwelling of global action" before climate change's impacts are irreversible.

*Planetary Boundaries Emergency.* An even more stark sense of urgency than the above two themes arises from the framing of the situation as a planetary emergency, for which one TC calls for "an emergency response". Another states the fundamental issue that TCs are trying to collectively deal with succinctly, "too often, these interconnected emergencies are viewed in silos when there is an urgent need to address them as one integrated challenge". Indeed, it is that very need that has given rise to the emergence of TCs in the first place.

Notably, the idea of urgency and huge system problems need to be supported by taking a systems orientation, rather than the more atomistic or fragmented approaches that many policymakers or even change agents sometimes take. This is where the final category of systems orientation—or systems thinking—becomes important for TC's catalytic action.

*4.4. Systems Orientation: The "Approach" to Transformation Catalysis*

The systems orientation of TCs or the approach of TC action is defined by adopting a systems orientation or understanding (systems thinking), which means thinking in terms of complex adaptive systems and wicked problems (with or without that specific language) and taking a holistic perspective on systemic change. More specifically, the systems orientation has the following three components: complex wickedness (wicked complexity), a holistic perspective, and a long-term orientation, as discussed below. Supplementary Material S5 and the synthesis in Table 1 show an orientation towards systems change/transformation around economics and development, as well as organizing to meet needs over the long term, through an emergent, complexity-based approach in this set of activities. Rather than being specific activities, in fact, this code articulates a systemic approach that understands the systemic nature of the problems being dealt with and approaches them with that understanding being apparent.

4.4.1. Complex Wickedness (Wicked Complexity)

The complexity wickedness (complex wickedness or wicked complexity) perspective acknowledges that the transformation of systems takes place in the context of complex, emergent adaptive systems fraught with wicked problems [1,7,10,41,42]. As the most common words suggest (Supplementary Material S5), complex wickedness recognizes interdependence (among people, in communities, and human to nature), argues for the co-creative emergence of ideas, solutions, and actions, sees self-similar units or fractals as part of many systems, acknowledges that healthy socio-economic (and ecological) systems self-organize and that there are many leverage points that have the potential to create tipping points for change. Complex wickedness has the following five themes, all related to ideas about complexity and wickedness: finding leverage points for change; emergence, self-organization, and fractals; interdependence and relationality; collaboration and co-creation; and self-organizing, as discussed below.

*Finding Leverage Points for Change*. TCs seem to draw from Meadows' [14] work on leverage points as target places for system change. One, for example, notes, "leverage points are places within complex system (a corporation, an economy, a living body, a city, an ecosystem) where a small shift in one thing can produce big changes in everything. By studying the systems in which we are acting, we can identify these leverage points and—critically—how we want to change them".

Key terms found in complexity and wicked problems theory pepper this code. Such terms include emergence, co-creation, collaboration, relationship, interdependence, even the idea of fractals (self-similar units at different levels of analysis), and self-organizing. Although it is difficult to parse out these terms separately, because they are often used with each other, below we provide insight into how different TCs express them.

*Emergence, Self-Organization, and Fractals*: As one TC puts it, the work of transformation is "emergent work: i.e., we don't know the answer in advance". Another put this idea in the following way: "emergence is how transformation happens in nature...the small is the large, . . . the microcosm actually IS the whole. The fractal is where the different aspects of transformation come together...AND therefore how we think about scale". Key to these ideas is recognition of the idea of self-organization. One TC creates "effective structure to guide that self-organizing". Another TC discusses the importance of "intention when we cluster, intention when we built trust, intention when we bridge...learn...take action...all in service of increasing the probability that the emergent outcomes will be consistent with our shared aspiration of a world where everyone belongs". Self-organization helps TCs "deal with complex, uncertain, and interconnected systems that are ever-changing", as they

"embrace a systems mindset". One of the TCs describes itself as a "complex network . . . that demonstrates a strong culture of self-organising".

*Collaboration and co-creation:* Critical to dealing with complex wickedness are collaborative and co-creative processes, which many TCs acknowledge explicitly, as follows: "co-define the state of the art"; "co-create the fractal" (or self-similar units at different levels of analysis); and "applying our collective learning". The idea of co-creation, as expressed by another TC, is linked to responsibility, as follows: "co-creation: we are each responsible for creating value; you get what you give". The idea that people everywhere have an agency that needs to be and can be tapped is also part of this way of thinking, as expressed by the following: "development initiatives must be interwoven with people and places, not imposed upon them".

*Interdependence and relationality:* Interdependence is another characteristic of complex wickedness—of initiatives and efforts and of the types of issues/problems that need to be resolved. One TC defines its orientation as to "define comprehensive solutions to the complex, interconnected challenges of our world", which are "manifesting in the multiple interconnected emergencies with which we are confronted today". As one TC puts it, "all levels of transformation are interdependent: change at any level of the system interacts with all other levels of the system". Alternatively, "think about how transformation happens, and how the levels of change . . . are interdependent". Another TC stated that "the process of transformation is inherently relational".

### 4.4.2. Holistic Perspective

Similar in some respects to complex wickedness, the holistic perspective argues for recognizing that everything in complex wickedness is interconnected, spans multiple levels and sectors, and therefore needs to be tackled holistically rather than in silos, because the systems of interest can be considered to be living systems. The top words used in this code (see Supplementary Material S5 and the synthesis in Table 1) emphasize changing or transforming systems, work, organizations, and the world (global) holistically and at scale. There are the following two aspects or themes to the holistic perspective: interconnectivity and a whole-system/entity orientation.

*Interconnectivity.* The idea of connectedness is common in how TCs understand their work, and it really embodies what it means to take a systems perspective. One TC expresses this notion clearly, "everything everywhere is linked in a single system, therefore, every action must be considered in the context of its effect on the whole system". Another takes that idea to the planetary level, stating the need to "recognize the interdependence of healthy people, planet and economies". This idea of interconnectivity can also be expressed in terms of catalytic action, as another TC does in this statement, "local transformation: how successful small-scale, local sustainability projects and new, evolving technologies can be made into successful mainstream initiatives. Global transformation: to catalyse strong and sustainable partnerships among people, communities, governments, NGO's and businesses on a global scale". The idea of interconnectivity, also part of complex wickedness in a sense, gives rise to the notion that "we are one" or, as one TC claims, "how the levels of change (I, We, World) are interdependent".

*Whole-System/Entity Orientation.* The theme of whole-system/whole entity orientation is clearly related to the notion of interconnectivity, but it takes different forms. One TC calls for "holistic economic thought that draws on the latest science of living systems, global wisdom", and another wants to "support holistic systems change interventions". The related goal of all of these initiatives is captured by one TC's statement that "we're working towards a vision of sustainable, equitable economies that work for all", where "all" tends to mean all people and all life for many of the TCs (though not universally, some just focus on people, while others, as noted above, take a more ecological stance).

### 4.4.3. Long-Term Orientation

The items coded into long-term orientation reflected seeing things and the prospects of systems over the long term, seeking long-lasting solutions and changes in systems to achieve long-term environmental and social sustainability for the future (see Supplementary Material S5 and synthesis in Table 1). This long-term orientation applies to both the initiatives themselves and to the actions that they work towards.

In this category, where there do not appear to be subthemes, one TC says that it will "support holistic systems change interventions, which require flexible support over the long term". Others have similar themes oriented toward long-term sustainability, with one typical example stating "enable the necessary change in systems to ensure long-term environmental and social sustainability". One TC wants to "safeguard [the] future" and another emphasizes both "short-and long-term systemic changes". The long-term orientation was well synthesized by one of the TCs, which stated that it would "harness our collective fears, build hope and drive action to respond to the human health, economic, climate and biodiversity crisis with solutions that build resilient societies in the longer-term".

## 5. Discussion and Conclusions

This paper has explored, in depth, the actions associated with the emerging organizational entity, called the transformation catalyst (TC), using data from their websites. Transformation catalysts represent an emerging way of organizing change agents and initiatives that we argue are needed to integrate transformation initiatives, so that they can more effectively address the complexity, interconnectedness, and wickedness of achieving transformational goals, such as the SDGs and other such efforts. As Table 1 illustrates (and the Supplementary Materials show in detail), the catalytic actions undertaken by TCs take many forms. The data indicate that system transformation, as defined by TCs, means dealing with whole systems and systemic solutions aimed at bringing about fundamental changes, through working to shift cognitive frames that reorient the mindsets and paradigms that help to explain a given system. It also means emphasizing systemic change—change of whole systems—through integrated approaches, shifting the core understanding about the place of human beings in the world, with respect to nature, working specific domains of change, and ensuring that all voices can be heard.

As defined by the TCs we studied, catalytic actions emphasize the "who" and "how" of bringing collaborators, partners, and allies together in new ways. This connects works across boundaries, so that allies can begin collaborating, coordinating, and co-creating initiatives oriented towards systemic change. TCs also work to help their allies cohere their efforts in a variety of ways, including developing insight through new learning and understandings, engaging in political activism and policy change, and creating dialogues that help to foster new types of alliances and actions. Amplifying action means strengthening and empowering diverse groups so that they can mobilize to take more effective action. This amplification process is aimed at creating ripples of new actions that result from initial actions, building coalitions, and strengthening a variety of capacities, e.g., for leadership, for action-taking, and for understanding the system, among others.

The sensemaking activity of TCs addresses the "when" and "why" of the need to take catalytic action, particularly around complex and interconnected issues, such as the ones embedded in the SDGs. Sensemaking means acknowledging the reason for (why) system transformation—and the urgency associated with that. That process involves problematizing specific domains, topics, or ways of doing things that are creating systemic problems, for example, ecological, economic, or inequality issues. In turn, problematizing creates a sense of urgency around the transformative impulse, engaging targeted actions in socio-economic, civic and political, and biogeophysical (planetary boundary) domains, and presumably in the future, others.

Accomplishing all of this change means adopting a systemic orientation that includes systems thinking, almost as a matter of conceptualization of the problem. The work of TCs makes it increasingly clear that systemic change of the sort envisioned by SDGs cannot

happen without the integration of numerous initiatives aiming at similar goals—and demands a systems approach. The language of complexity and wicked problems thus finds its way into many descriptions of what TCs are doing, e.g., words such as leverage points, emergence, interdependence, and relationality, along with co-creation and self-organization, which are common descriptors. In addition, taking a long-term orientation is a given for many TCs, who assume a holistic approach to the issue or problem domain that recognizes interconnectivity and takes a whole system, rather than piecemeal, orientation.

An understanding of what TCs are and how they operate is timely and important, because wickedly complex challenges associated with the UN SDGs require transformational change, not incremental or piecemeal approaches. Such transformational changes cannot be achieved by any single entity alone, be it government agencies, businesses, or NGOs. What is needed is the collective and coherent action of many initiatives guided by common aspirations. We believe that TCs may be a ray of hope in this context of complexly wicked problems. Moreover, by analyzing how the entities identified as TCs actually undertake their work and identify commonalities across such entities, this study provides those interested in system transformations with an understanding of what elements are needed if TCs are to work, and how new ones can emerge.

This research has limitations. As we mentioned earlier, our paper is exploratory in nature and the 27 organizations that we identified in this paper are by no means an exhaustive list of TCs. Building on the criteria that we developed in this paper, future studies could look more into other existing TCs and see if there are other ways that TCs are distinctive from other types of organizations, and gain more insight into the domains in which they are active, how they actually operate day-to-day, and, importantly, what their actual impact on systemic transformation is. Moreover, because our study relied mostly on website materials, it captures TCs' rhetoric about themselves and their work, which may or may not overlap with what they actually do in practice, or how effective their efforts actually are, which can certainly be addressed in future research, e.g., through case analysis or participant observation. For the purpose of this paper, and also because most TCs that we identified are relatively new or in their early stages, we did not (and were not able to) collect detailed data on their hands-on practices. Thus, much more research is needed that looks at the actual practices of TCs in more detail and documents how TCs differ in how they work from the other catalytic entities discussed in the literature review section.

**Supplementary Materials:** The following are available online at https://www.mdpi.com/article/10.3390/su13179813/s1, Supplementary Material S1. Criteria for Transformation Catalysts, Supplementary Material S2. Transformations Agenda: The What and Where of Transformation Catalysts' Work, Supplementary Material S3. Catalytic Action: The Who and How of Transformations Catalysts' Work, Supplementary Material S4. Sensemaking: The Why and When of Catalyst Transformations' Work, Supplementary Material S5. Systems Orientation: The Approach to Transformation Catalysts' Work.

**Author Contributions:** All authors contributed equally to the data gathering, analysis, and writing of this manuscript in an iterative process. All authors have read and agreed to the published version of the manuscript.

**Funding:** No funding was received for this research.

**Data Availability Statement:** The authors will provide a list of websites studied on request. All data are (were) publicly available at the time of data gathering.

**Acknowledgments:** The authors would like to acknowledge the Bounce beyond Steward Team, which has been working on related issues.

**Conflicts of Interest:** The authors declare no conflict of interest.

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
