# Peer review of "How Transformation Catalysts Take Catalytic Action"

_sustainability, doi:10.3390/su13179813_

Round 1

Reviewer 1 Report

I very much enjoyed reading this excellent article and its exploration of the potential value and contribution of TCs towards addressing the challenges of implementing the SDGs and achieving related targets.

I liked the inclusion of innovation brokers or intermediaries in your review of Antecedents of Transformation Catalysts, but was surprised that there was no mention of any of the literature on partnership brokers, for example, Hundal (2013) 'The role partnership brokers play in creating effective social partnerships' (in Seitanidi & Crane's Social Partnerships and Responsible Business (Routledge) and Stott (2019) Shaping Sustainable Change: The Role of Partnership Brokering in Optimising Collaborative Action (Routledge) and Gehringer (2020) 'Corporate Foundations as Partnership Brokers in Supporting the United Nations’ Sustainable Development Goals (SDGs)'. There are also extensive practitioner texts and resources on partnership brokering such as Tennyson (2005) The Brokering Guidebook (The Partnering Initiative) and Mundy & Tennyson (2019) Brokering Better Partnerships (Partnership Brokers Association). And PBA's Betwixt & Between – The Journal of Partnership Brokering.

I have noted a few minor things that I think need to be edited as follows:

Line 7: the 17 United Nations Sustainability Sustainable Development Goals (SDGs)

Lines 18-19: (3) TCs clearly acknowledge the current status quo, attributions, and urgency (i.e., sensemaking), and finally, (4) TCs embody systems orientation.

Author Response

Thank you so much for this comment. We’ve added partnership brokers in our discussion of antecedents and also elaborated how TCs are also distinct from partnership brokers. In short, we note that much of the partnership broker literature is not transformative, which is what TCs are attempting.

Reviewer 2 Report

In the paper presented for review, the authors attempt to thoroughly analyze the catalytic activities of the new form of organization, which can potentially facilitate the timely implementation of transformation aspirations, such as the SDGs: Transformation Catalyst (TC).

The innovative nature of the issues and extensive considerations deserve attention.

The structure of the reviewed article is well-thought-out and transparent.

The article definitely meets the requirements of the scientific text. Formally and substantially, it is flawless.

Author Response

Thank you so much for your positive feedback! As there were no specific comments, we simply thank you. 

Reviewer 3 Report

The article is of a very narrow theoretical nature, it may not be of interest to a wide range of scientists. I think that it is necessary to pay attention to the practical part, which is completely absent in the article — the analysis of statistics, the development of strategies for the analyzed 27 organizations, etc. Also, it is necessary to point out the following notes:

  1. The novelty of the author's study is not highlighted. Now this is only an analysis of the methodological platform, in terms of content it is not an original scientific article, but a review.
  2. Abstract must be built as follows: purpose of the article, methods, results, conclusions, and recommendations/future directions. Now the elements of novelty, what is done by the author, are not clear.
  3. Introduction is not built properly. The relevance of the study has not been proven, there are no objectives and hypotheses of the study.
  4. In the article and its Appendices, there are a lot of tables without a description. The article is overloaded with tables, there are a lot of repeated information.
  5. It is necessary to change the structure of the article: Results and Discussion, and separately Now Discussion is absent at all — compare your results with those of other authors. In Conclusions, it is not highlighted what specifically was done by the authors, highlight the prospects for further studies.
  6. In the References section, it is necessary to check the sources — the subject matter of publications does not correspond to the subject of the study. For example, 3 citations of Klerkx, L.
  7. Problems with article design. For example, references in the text are incorrectly formatted — just quotation marks without links to publications (line 645-674).

Author Response

Responses to Reviewer 3 [Reviewer Comments are Italicized]

  1. The novelty of the author's study is not highlighted. Now this is only an analysis of the methodological platform, in terms of content it is not an original scientific article, but a review. Abstract must be built as follows: purpose of the article, methods, results, conclusions, and recommendations/future directions. Now the elements of novelty, what is done by the author, are not clear.

We believe that the novelty of study comes from the fact that we are introducing a novel form of organizing (i.e., TCs) which can act as transformation agents especially when dealing with complex problems such as the SDGs. The paper also focuses on what is potentially a more effective way to bring system change about—through catalytic actions by these emerging entities. Our paper is the first empirical study of TCs that have the kind of transformative potential that we know of. We believe that the notion of TCs and their distinct characteristics, if understood correctly, can be useful for many organizations that are working towards bringing about transformation. In order to emphasize this point more clearly, we’ve also added a part in the introduction where we elaborate more fully how we think our study contributes to the literature as well as to the timely achievement of SDGs.

  1. Introduction is not built properly. The relevance of the study has not been proven, there are no objectives and hypotheses of the study.

As we mentioned in the manuscript, our paper is exploratory in nature and descriptive in its results. Our objective was to describe and analyze the catalytic actions of a novel form of organizing, what we call the Transformation Catalysts (TCs), that could potentially facilitate the timely achievement of transformational aspirations like the SDGs. Since our paper does not adopt quantitative approaches to data analyses, we do not have any hypothesis.

  1. In the article and its Appendices, there are a lot of tables without a description. The article is overloaded with tables, there are a lot of repeated information.

Thank you for the comment. We agree with you that the tables were a bit too overwhelming with redundant information. Therefore, following your feedback, we took out the Appendices from the manuscript and have posted them as online Appendix (with link included) so that people who would like a more detailed information could take a look.

  1. It is necessary to change the structure of the article: Results and Discussion, and separately Now Discussion is absent at all — compare your results with those of other authors. In Conclusions, it is not highlighted what specifically was done by the authors, highlight the prospects for further studies.

Thank you for your comment. The paper does contain separate sections on results, limitations (to which we added Implications for Future Research), and a combined discussion and conclusion section that synthesizes our core findings.

  1. In the References section, it is necessary to check the sources — the subject matter of publications does not correspond to the subject of the study. For example, 3 citations of Klerkx, L.

We are sorry for the confusion. We are citing three different studies by Kelrkx, as we believe that they are all relevant to our study as they talk about innovation brokers or innovation intermediaries that share some features with transformational catalysts.

  1. Problems with article design. For example, references in the text are incorrectly formatted — just quotation marks without links to publications (line 645-674).

The quotation marks are used because they are direct quotes from our (website) data that are meant to illustrate the findings and what different TCs are saying specifically. They are not from academic journal articles and therefore citations are not needed, as we have chosen to keep the particular initiatives studied anonymous for purposes of this research.

Round 2

Reviewer 3 Report

After another review, I have to admit that the authors of the article did not make any corrections according to my comments. In their responses, the authors unreasonably make an attempt to defend their opinion. So, I duplicate all the comments from Round 1 of the review — nothing has been corrected.

  1. The novelty of the author's study is not highlighted. Now this is only an analysis of the methodological platform, in terms of content it is not an original scientific article, but a review.
  2. Abstract must be built as follows: purpose of the article, methods, results, conclusions, and recommendations/future directions. Now the elements of novelty, what is done by the author, are not clear.
  3. Introduction is not built properly. The relevance of the study has not been proven, there are no objectives and hypotheses of the study.
  4. In the article and its Appendices, there are a lot of tables without a description. The article is overloaded with tables, there are a lot of repeated information.
  5. It is necessary to change the structure of the article: Results and Discussion, and separately Now Discussion is absent at all — compare your results with those of other authors. In Conclusions, it is not highlighted what specifically was done by the authors, highlight the prospects for further studies.
  6. In the References section, it is necessary to check the sources — the subject matter of publications does not correspond to the subject of the study. For example, 3 citations of Klerkx, L.
  7. Problems with article design. For example, references in the text are incorrectly formatted — just quotation marks without links to publications (p. 645-674).

Author Response

Thank you. 

We have carefully proofed the manuscript, added comments about the significance of the work, and added in information in the abstract, and shifted the implications per your request. We thank you for your attention to the manuscript.